# Prognostic and Predictive Role of Body Composition in Metastatic Neuroendocrine Tumor Patients Treated with Everolimus: A Real-World Data Analysis

**DOI:** 10.3390/cancers14133231

**Published:** 2022-06-30

**Authors:** Nicoletta Ranallo, Andrea Prochoswski Iamurri, Flavia Foca, Chiara Liverani, Alessandro De Vita, Laura Mercatali, Chiara Calabrese, Chiara Spadazzi, Carlo Fabbri, Davide Cavaliere, Riccardo Galassi, Stefano Severi, Maddalena Sansovini, Andreas Tartaglia, Federica Pieri, Laura Crudi, David Bianchini, Domenico Barone, Giovanni Martinelli, Giovanni Luca Frassineti, Toni Ibrahim, Luana Calabrò, Rossana Berardi, Alberto Bongiovanni

**Affiliations:** 1Osteoncology and Rare Tumors Center, IRCCS Istituto Romagnolo per lo Studio dei Tumori (IRST) “Dino Amadori”, 47014 Meldola, Italy; nicoletta.ranallo@irst.emr.it (N.R.); luana.calabro@irst.emr.it (L.C.); 2Radiology Unit, IRCCS Istituto Romagnolo per lo Studio dei Tumori (IRST) “Dino Amadori”, 47014 Meldola, Italy; andrea.prochowskiiamurri@irst.emr.it (A.P.I.); domenico.barone@irst.emr.it (D.B.); 3Unit of Biostatistics and Clinical Trials, IRCCS Istituto Romagnolo per lo Studio dei Tumori (IRST) “Dino Amadori”, 47014 Meldola, Italy; flavia.foca@irst.emr.it; 4Osteoncology Unit, Bioscience Laboratory, IRCCS Istituto Romagnolo per lo Studio dei Tumori (IRST) “Dino Amadori”, 47014 Meldola, Italy; chiara.liverani@irst.emr.it (C.L.); alessandro.devita@irst.emr.it (A.D.V.); laura.mercatali@irst.emr.it (L.M.); chiara.calabrese@irst.emr.it (C.C.); chiara.spadazzi@irst.emr.it (C.S.); 5Unit of Gastroenterology and Digestive Endoscopy, Forli-Cesena Hospital, AUSL Romagna, Cesena, 47121 Forli, Italy; carlo.fabbri@auslromagna.it; 6General and Oncologic Surgery Unit, Morgagni-Pierantoni Hospital, AUSL Romagna, 47121 Forlì, Italy; davide.cavaliere@auslromagna.it; 7Nuclear Medicine and Radiometabolic Unit, IRCCS Istituto Romagnolo per lo Studio dei Tumori (IRST) “Dino Amadori”, 47014 Meldola, Italy; riccardo.galassi@auslromagna.it (R.G.); stefano.severi@irst.emr.it (S.S.); maddalena.sansovini@irst.emr.it (M.S.); 8Endocrinology Unit, Forli-Cesena Hospital, AUSL Romagna, Cesena, 47121 Forli, Italy; andreas.tartaglia@auslromagna.it; 9Pathology Unit, Morgagni-Pierantoni Hospital, AUSL Romagna, Cesena, 47121 Forli, Italy; federica.pieri@auslromagna.it; 10Oncology Pharmacy Unit, IRCCS Istituto Romagnolo per lo Studio dei Tumori “Dino Amadori”, 47014 Meldola, Italy; laura.crudi@irst.emr.it; 11Medical Physics Unit, Istituto Scientifico Romagnolo per lo Studio e la Cura dei Tumori (IRST) IRCCS, 47014 Meldola, Italy; david.bianchini@irst.emr.it; 12IRCCS Istituto Romagnolo per lo Studio dei Tumori “Dino Amadori”, 47014 Meldola, Italy; giovanni.martinelli@irst.emr.it; 13Department of Medical Oncology, IRCCS Istituto Romagnolo per lo Studio dei Tumori (IRST) “Dino Amadori”, 47014 Meldola, Italy; luca.frassineti@irst.emr.it; 14Osteoncology, Bone and Soft Tissue Sarcomas and Innovative Therapies Unit, IRCCS Istituto Ortopedico Rizzoli, 40136 Bologna, Italy; toni.ibrahim@ior.it; 15Department of Medical Oncology, Università Politecnica delle Marche, AOU Ospedali Riuniti di Ancona, 60126 Ancona, Italy; r.berardi@staff.univpm.it

**Keywords:** NET, mTOR inhibitors, everolimus, body composition, muscle tissue, adipose tissue, neuroendocrine tumors

## Abstract

**Simple Summary:**

Neuroendocrine tumors (NETs) are rare neoplasms that often present upregulation of the mammalian rapamycin targeting pathway (mTOR) with consequent uncontrolled growth and proliferation. This pathway is also involved in the metabolism of adipose tissue and in the regulation of skeletal muscle synthesis. The mTOR therefore represents an attractive therapeutic target. Everolimus acts by selectively inhibiting the mTOR pathway with an antiproliferative effect. The aim of this study is to investigate the prognostic and predictive role of body composition indices (muscle and adipose) in metastatic NETs patients treated with everolimus.

**Abstract:**

Neuroendocrine tumors (NETs) are rare neoplasms frequently characterized by an upregulation of the mammalian rapamycin targeting (mTOR) pathway resulting in uncontrolled cell proliferation. The mTOR pathway is also involved in skeletal muscle protein synthesis and in adipose tissue metabolism. Everolimus inhibits the mTOR pathway, resulting in blockade of cell growth and tumor progression. The aim of this study is to investigate the role of body composition indexes in patients with metastatic NETs treated with everolimus. The study population included 30 patients with well-differentiated (G1-G2), metastatic NETs treated with everolimus at the IRCCS Romagnolo Institute for the Study of Tumors (IRST) “Dino Amadori”, Meldola (FC), Italy. The body composition indexes (skeletal muscle index [SMI] and adipose tissue indexes) were assessed by measuring on a computed tomography (CT) scan the cross-sectional area at L3 at baseline and at the first radiological assessment after the start of treatment. The body mass index (BMI) was assessed at baseline. The median progression-free survival (PFS) was 8.9 months (95% confidence interval [CI]: 3.4–13.7 months). The PFS stratified by tertiles was 3.2 months (95% CI: 0.9–10.1 months) in patients with low SMI (tertile 1), 14.2 months (95% CI: 2.3 months-not estimable [NE]) in patients with intermediate SMI (tertile 2), and 9.1 months (95% CI: 2.7 months-NE) in patients with high SMI (tertile 3) (*p* = 0.039). Similarly, the other body composition indexes also showed a statistically significant difference in the three groups on the basis of tertiles. The median PFS was 3.2 months (95% CI: 0.9–6.7 months) in underweight patients (BMI ≤ 18.49 kg/m^2^) and 10.1 months (95% CI: 3.7–28.4 months) in normal-weight patients (*p* = 0.011). There were no significant differences in terms of overall survival. The study showed a correlation between PFS and the body composition indexes in patients with NETs treated with everolimus, underlining the role of adipose and muscle tissue in these patients.

## 1. Introduction

Neuroendocrine neoplasms (NENs) represent a heterogeneous group of tumors. The incidence and prevalence have significantly increased in recent decades; however, they are still classified as rare diseases, with a global incidence that is fewer than six new cases per year for every 100,000 individuals [1,2,3].

NENs are generally sporadic, but associations with genetic syndromes are described. The clinical presentation is extremely heterogeneous. The well-differentiated subgroups characterized by a good prognosis are defined as Neuroendocrine Tumors (NETs) [4,5].

Several treatments for NETs have been validated or investigated in prospective clinical trials. These have focused on an anti-proliferative effect and include somatostatin analogues (SSAs) or multi-kinase inhibitors, such as sunitinib, lenvatinib, and pazopanib; and the mammalian target of rapamycin (mTOR) inhibitor everolimus [6,7].

mTOR is a serine-threonine kinase that coordinates and transduces signals from various growth factors and upstream proteins; its pathway controls the regulation of metabolism, proliferation, and cell survival. mTOR forms at least two distinct, functional multi-protein complexes [8]. Generally, mTOR complex 1 (mTORC1) controls autonomous cell growth, whereas mTORcomplex 2 (mTORC2) regulates cell proliferation and survival [9]. In normal cells, mTOR controls cell function and homeostasis. Conversely, in tumor cells, mTOR undergoes hyperactivation, resulting in uncontrolled proliferation and tumor growth [10,11,12,13].

Everolimus acts as a selective inhibitor of the mTOR pathway; it acts on binding to an intracellular protein, FKBP-12, and forms a complex that inhibits mTORC1 activity. The inhibition of the mTORC1 signaling pathway interferes with the transduction and synthesis of proteins involved in the cell cycle, angiogenesis, and glycolysis. Everolimus therefore inhibits the growth and proliferation of tumor cells, endothelial cells, fibroblasts, and smooth muscle cells associated with blood vessels [14,15,16,17,18] (Figure 1).

mTOR also plays a key role in the regulation of lipid and carbohydrate metabolism [19], and it is also involved in the metabolism and regulation of adipose tissue [20]. Adipocytes play a critical role in modulating the microenvironment to which cancer cells are exposed. Fat cells secrete inflammatory growth factors and cytokines; among these, leptin and interleukin 6 (IL-6) play roles in the activation of the mTOR signaling pathway and are involved in tumor progression and resistance to chemotherapy treatments [21].

Furthermore, the target phosphoinositide 3-kinase (PI3K)/protein kinase B (Akt)/rapamycin (mTOR) pathway also plays a key role in activating skeletal muscle synthesis. The upregulation of this pathway leads to muscle hypertrophy, and the genetic block determines a block on the hypertrophy [22]. The use of sorafenib, another multi-kinase inhibitor, has also been related to the onset of sarcopenia, probably as a result of the downstream suppression of PI3K, Akt, and mTOR [23].

Several studies have suggested that sarcopenia is independently associated with a worse prognosis in patients with cancer [24,25]. Sarcopenia predicts survival regardless of body weight; moreover, a reduced muscle mass is not only observed in cachectic individuals but also in overweight or patients [26]. This “sarcopenic obesity” is correlated with reduced survival and complicates the diagnosis of sarcopenia in patients suffering from diseases such as oncological diseases.

These findings underline the importance of quantifying muscle and fat percentage, as opposed to just evaluating total weight and body mass index (BMI) [24,27].

An indicator of muscle mass is the skeletal muscle index (SMI), which is calculated using computed tomography (CT). A CT scan is an excellent tool for quantifying muscle mass and for measuring the different deposits of adipose tissue [28].

SMI seems to be a prognostic factor for overall survival (OS) in patients treated for gastrointestinal tract cancer; OS differs between patients with and without pre-sarcopenia (Hazard ratio = 1.92, CI 1.02–3.60, *p* = 0.043) [29].

The aim of this study is to evaluate the role of the visceral and subcutaneous adiposity index and the SMI in a real-world population with metastatic NET treated with everolimus, and in particular to investigate whether these indexes can predict outcome or response to treatment.

## 2. Patients and Methods

### 2.1. Population

From 1 November 2013 to 31 August 2021, data on 49 patients with locally advanced (unresectable) or metastatic NET who had received treatment with everolimus were retrospectively collected from a database available at the IRCCS Romagnolo Institute for the Study of Tumors (IRST) “Dino Amadori”, Meldola (FC), Italy, which is a member of the European Union Reference Network for Rare Cancers Neuroendocrine Tumor Group (EURACAN G4 NET).

Thirty patients who met the inclusion criteria and had undergone a baseline radiological study with a CT scan before the start of treatment with everolimus (at a dose of 10 mg/day every 28 days) and at the first radiological evaluation after approximately 3 months of everolimus therapy were considered eligible for the study. Nineteen patients were excluded because of the unavailability of imaging from the radiological archive (Picture Archiving and Communication System) (Figure 2).

Patient and tumor characteristics (histology, grading, and secretory status), prior treatments, and functional imaging properties acquired from the CT scan and somatostatin receptor imaging (Ga68 DOTATOC/DOTATATE PET/CT) were retrospectively collected.

Based on a previous published study [30], patients were also divided into two groups according to the Ki-67% (≤10% and >10%).

All patients had regular clinical and radiological follow-ups of CT, MRI, 68Ga-PET, and/or 18-fluorodeoxyglucose (FDG)-PET; the radiological information relating to the response to treatments was collected up to the death of the patient or until the last follow-up.

The decision to start everolimus-based therapy was made by a multidisciplinary specialized tumor board at our institute. Everolimus was used as a second, third, or subsequent line of treatment and was administered at a dosage of 10 mg/day every 28 days. Treatment was stopped in cases of disease progression, unacceptable toxicity, or death.

The main adverse effects were classified according to the National Cancer Institute Common Terminology Criteria for Adverse Events version 5.0 [31]. Response to treatment was defined according to the radiological criteria in RECIST version 1.1 [32].

### 2.2. Calculation of Body Composition Indexes

The CT images were acquired with similar acquisition parameters: tube voltage between 100 and 120 kVp, automatic tube current, soft tissue reconstruction algorithm, a 512 × 512 matrix, and a slice thickness of 5 mm. From these acquisitions, a single CT image localized at the third lumbar vertebral body (L3), depicting both the transverse processes, was selected by an experienced radiologist. The image was processed with ABACS software (version 2.0, Voronoi Health Analytics, Vancouver, BC, Canada) that, in a fully automated fashion, segmented and determined the cross-sectional surfaces of skeletal muscles (SKM, including paraspinal muscles, abdominal wall muscles, and the psoas), subcutaneous adipose tissue (SAT, including intramuscular fat tissue), and visceral adipose tissue (VAT) (Figure 3) [33]. All the segmentations and the measurements were reviewed, corrected when necessary, and validated by the radiologist.

The obtained surfaces (*SKM*, *VAT*, and *SAT*) were then normalized for the square of the heights to obtain indexes (*SMI*, *VATI*, *SATI*) using the following formula: $$SMI;VATI;SATI=\frac{SKM;VAT;SAT\;\left({\mathrm{cm}}^{2}\right)}{Height^{2}\;(m^{2})}$$

The total adipose tissue index (*TATI*) derives from the sum of *SATI* and *VATI*.

The *BMI*, assessed before the start of everolimus therapy, was calculated using the following formula:$$BMI=\frac{Weight\;\left(\mathrm{Kg}\right)}{Height^{2}\;(m^{2})}$$

Weight and height were obtained from the medical record. The results obtained were divided into four categories: underweight (*BMI* ≤ 18.49 kg/m^2^), normal weight (*BMI* between 18.5 and 24.99 kg/m^2^), overweight (*BMI* between 25 and 29.99 kg/m^2^), and (*BMI* ≥ 30 kg/m^2^).

### 2.3. Statistical Analysis

Continuous variables were presented as median (min-max values), and categorical variables were presented as absolute and relative frequencies. Indexes were evaluated as continuous values and by using tertiles. To test the equality of matched pairs of observations among pre-therapy and post-therapy assessments, the Wilcoxon matched-pairs signed-rank test was used. To evaluate the pre-therapy index according to the best response, the Wilcoxon rank-sum test was used. The chi-squared test was used to evaluate the variation in best response from pre-therapy to post-therapy among different subgroups and to analyze the toxicities between different tertiles. OS was calculated from the first day of treatment to the day of the patient’s death or last date of visit. Progression-free survival (PFS) was calculated from the first day of treatment to the day of the progressive disease or last date of assessment. The best response was defined as the best response recorded from the start of treatment with everolimus until death or disease progression occurred. Time-to-event data were described using Kaplan–Meier curves, and different subgroups of patients were compared with the log-rank test. Ninety-five percent confidence intervals (95% CIs) were calculated by non-parametric methods. Data on pre- and post-everolimus-based therapy indexes were graphically summarized using scatterplots. Statistical analyses were carried out using STATA/MP 15.0 for Windows (StataCorpLLC, College Station, TX, USA).

### 2.4. Statement of Ethics

The retrospective study was conducted according to the ethical standards established in the Helsinki Declaration of 1964 and was approved by the IRST IRCCS Ethics Committee (project identification code: #L1P33). Informed consent was not required because of the retrospective nature of the study and the use of anonymized clinical data.

## 3. Results

### 3.1. Patient Characteristics

This retrospective analysis was performed on data from 30 patients with metastatic NETs (G1-G2) who were treated with everolimus. The population was equally divided between men and women. More than 70% of patients had a diagnosis of gastroenteropancreatic (GEP) NET (41.4% gastrointestinal and 31.0% pancreatic). Approximately 80% of patients had a G2 NET, and 64% showed a Ki-67 ≤ 10%.

Baseline BMI was normal (between 18.50 and 24.99 kg/m^2^) for the 60% of patients, but 26.7% of patients were underweight (BMI ≤ 18.49 kg/m^2^), and the remaining 13.3% were overweight (BMI ≥ 25 kg/m^2^). Table 1 summarizes the characteristics of the patients included in the study.

### 3.2. Correlation between Body Composition and Outcome

Median follow up was 36.1 months (range: 4.2–83.8). Median PFS was 8.9 months (95% CI: 3.4–13.7 months), and median OS was 34.5 (95% CI: 14.6-NE months). Twenty patients (66.7%) obtained Disease Control Rate (DCR) with a partial response (PR) in 13.3% of cases or stable disease (SD) in 53.4% of cases as the best response, whereas 10 patients (33.3%) experienced disease progression (PD) at the first radiological evaluation.

The analysis of the muscle and fat index variation, measured before and after the start of everolimus-based treatment, showed a statistically significant difference in the SMI (median values, respectively, of 42.26 cm^2^/m^2^ vs. 39.95 cm^2^/m^2^; *p* = 0.005), the SATI (median values, respectively, of 50.01 cm^2^/m^2^ vs. 46.42 cm^2^/m^2^; *p* < 0.001), and the TATI (median values, respectively, of 105.02 cm^2^/m^2^ vs. 46.42 cm^2^/m^2^; *p* < 0.001). No difference was recorded in the VATI index before versus after the start of everolimus therapy (median values, respectively, of 31.65 cm^2^/m^2^ vs. 30.63 cm^2^/m^2^; *p* = 0.338; Figure 4).

The relationship between the best response and the body composition parameters at baseline did not demonstrate a significant difference in SMI between patients showing SD or PR and patients with PD (respectively, 42.5 cm^2^/m^2^ [range: 32.8–64.9 cm^2^/m^2^] vs. 39.2 cm^2^/m^2^ [range: 31.0–46.6 cm^2^/m^2^]; *p* = 0.186).

Conversely, the abdominal adipose tissue indexes (SATI, VATI, and TATI) were significantly higher at baseline in patients with SD or PR compared with patients experiencing PD: 66.6 cm^2^/m^2^ (range: 13.4–154.2 cm^2^/m^2^) versus 41.3 cm^2^/m^2^ (range: 14.2–103.4) for SATI (*p* = 0.086), 40.2 cm^2^/m^2^ (range: 3.9–108.5 cm^2^/m^2^) versus 7.4 cm^2^/m^2^ (range: 2.8–76.8) for VATI (*p* = 0.015); and 122.4 cm^2^/m^2^ (range: 17.4–214.1 cm^2^/m^2^) versus 49.2 cm^2^/m^2^ (range: 17.0–136.4) for TATI (*p* = 0.027; Table 2).

We also evaluated the correlation between the variation of body composition parameters during everolimus therapy and the disease response (Table 3); 45% of patients with SD or PR showed an increased SMI after the start of therapy, whereas 11 patients (55%) with SD or PR and all the patients (100%) with early PD (*n* = 10) showed lower SMI values from baseline to the first disease radiological evaluation (*p* = 0.011). There was no correlation between the variation of the abdominal adipose tissue indexes and the response to therapy for VATI (*p* = 0.492), SATI (*p* = 1.000), or TATI (*p* = 0.150).

As shown in Table 4 and as illustrated in Figure 5, after patient data were stratified by tertiles, the PFS was 3.2 months (95% CI: 0.9–10.1 months) in patients with low SMI (tertile 1), 14.2 months (95% CI: 2.3 months-not estimable [NE]) in patients with intermediate SMI (tertile 2), and 9.1 months (95% CI: 2.7 months-NE) in patients with high SMI (tertile 3). The difference was statistically significant (*p* = 0.039). Similarly, the other body composition indices also showed a statistically significant difference in the three groups identified by tertiles.

When compared by BMI, the median PFS was significantly lower for underweight patients (BMI ≤ 18.49 kg/m^2^) than for normal-weight patients (3.2 months [95% CI: 0.9–6.7 months] vs. 10.1 months [95% CI: 3.7–28.4 months]; *p* = 0.011). There were no statistically significant differences in terms of age, gender, or Ki-67. The median OS was 34.5 months (95% CI: 14.6 months-NE). There were no significant differences in OS within the subgroups of tertiles for the different body composition indices. In particular, the OS was 14.6 months (95% CI: 6.0–44.4 months) in patients with low SMI (tertile 1), 17.5 months (95% CI: 4.1-NE months) in patients with intermediate SMI (tertile 2) and not reached (NR) in patients with high SMI (tertile 3) (*p* = 0.103). The same for the other body composition indexes (VATI, *p* = 0.167; SATI, *p* = 0.163 and TATI, *p* = 0.204).

### 3.3. Correlation between Body Composition and Toxicity

As shown in Table 5, 86.7% of patients showed at least one adverse effect (of any degree) during treatment with everolimus; 33% of patients (*n* = 10) showed at least one grade 3 toxicity, of which 16.7% was mucositis and 10% was liver function abnormalities (increase in aspartate aminotransferase (AST), alanine aminotransferase (ALT) and/or gamma-glutamyl transferase (GGT)).

There was no correlation between the onset of everolimus-related adverse effects and the different subgroups of patients stratified into tertiles for the body composition parameters, with the exception of the SATI (*p* = 0.024). Fifty percent of patients who experienced a grade 1–2 toxicity or no toxicity were assigned to tertile 1; 25% of these toxicities occurred in patients in tertile 2, and the remaining 25% occurred in patients in tertile 3. In contrast, none of the patients who reported grade 3 toxicity ranked in the first tertile for SATI; rather, these toxicity-reporting patients were equally divided between tertiles 2 and 3 (Table 6).

## 4. Discussion

NETs are rare malignancies arising from the neuroendocrine system (2). The mTOR pathway plays a key role in controlling the cell cycle and tumor growth. This pathway is frequently upregulated in these neoplasms representing an optimal therapeutic target [16,19].

Everolimus is an mTOR inhibitor approved for the treatment of advanced or metastatic NETs [14]. Recently, Barrea et al. [34] have shown that visceral adipose tissue and increased BMI are negative prognostic factors in patients with gastrointestinal and pancreatic neuroendocrine tumors.

mTOR is also involved in the regulation of anabolic and catabolic signaling of muscle mass, with consequent modulation of hypertrophy and muscle wasting [22,35].

The pathway of rapamycin (i.e., mTOR) plays a key role in activating skeletal muscle synthesis through various mechanisms. According to in vivo experiments, upregulation of this pathway leads to muscle hypertrophy, whereas inhibition causes hypotrophy. Therefore, blocking this pathway could lead to muscle loss in patients receiving an mTOR inhibitor [22,36].

Furthermore, a possible role of mTOR inhibitors in preventing and/or reversing tumor-associated cachexia through restoration of autophagy or reduction of IL-6 levels has also been demonstrated [37,38]. Hatakeyama et al. [38] showed that the benefit of everolimus in counteracting muscle atrophy can derive from a direct inhibition of muscle mTOR, but it also can be linked to its antitumor efficacy.

In our study, the analysis carried out on 30 patients with metastatic NETs who were treated with everolimus showed a correlation between the survival outcome and skeletal muscle and adipose indexes. Patients who showed an increase in the SMI value after starting treatment had a better response rate (SD or PR) than did patients with a decreased SMI value from baseline to the first radiological evaluation after initiation of everolimus therapy.

In addition, a statistically significant difference was noted between normal-weight patients (BMI between 18.5 and 24.99 kg/m^2^) and patients with a BMI ≤ 18.49 kg/m^2^ before starting treatment with everolimus. From the data obtained, it appears that patients with a higher BMI have better PFS during treatment than underweight patients. This is a very important finding as it highlights the negative prognostic role of a low BMI in patients with metastatic neuroendocrine tumors. From our knowledge, in fact, there is little data regarding the role of this index in patients with NETs.

The mechanism by which visceral obesity and increased body mass can improve the response to treatment is not well understood.

Patients with reduced muscle mass may express low levels of mTORC1 receptors compared with patients who have greater muscle mass; therefore, the effect of mTOR inhibition may be less evident.

There was a statistically better PFS in patients with increased visceral, subcutaneous, and total body fat indices. This benefit could be related to the ability of everolimus to inhibit the mTOR pathway, which is upregulated in adipose tissue.

Patients with increased subcutaneous and visceral adipose tissue may have greater mTOR activation. Furthermore, the inhibition of the mTOR signaling pathway would result in a lower production of growth factors and inflammatory cytokines involved in tumor progression; in these patients, the antitumor effect of everolimus could be more marked. This concept is only a hypothesis, and additional investigations are needed to study the biological mechanisms underlying any connection.

However, visceral adiposity remains an indicative parameter of nutritional status, so the exhaustion of body fat can also be a sign of neoplastic cachexia. Furthermore, malnutrition is associated with reduced survival in various neoplasms as well as reduced benefit from systemic treatments and increased treatment-related toxicity.

As regards toxicity, however, there does not seem to be a correlation between the degree of toxicity recorded in relation to tertiles for SMI, VATI and TATI; only the SATI index showed a significant difference, highlighting a possible correlation between the increase in subcutaneous adipose tissue and grade 3 toxicity.

Gyawali et al. [39] reported a significant correlation between SMI changes after mTOR inhibitors treatment compared with the baseline SMI; patients who exhibited low baseline muscle mass did not document the same muscle mass loss during treatment as patients who had greater baseline muscle mass. For the important role of the mTOR pathway in muscle synthesis, patients who have low baseline muscle mass may have low levels of mTORC1 receptors compared with patients who have relatively higher muscle mass. The authors hypothesized that the effect of mTOR inhibition may not be as pronounced in patients with low muscle mass [39].

Auclin et al. [40] also showed that SMI is an independent prognostic factor in a patient with renal cell carcinoma who was treated with everolimus. mTOR is also essential for adipogenesis and for the maintenance of fatty mass. Adipose tissue must not be classified only as a static site for energy storage but also must be considered a highly active endocrine and metabolic organ capable of regulating the immune response, blood pressure, angiogenesis, bone mass, and reproductive function [41].

Several studies have shown that mTOR signaling plays a crucial role in adipose functions, such as adipogenesis, lipid metabolism, and thermogenesis [42] (Figure 6). The tumor microenvironment is essential in growth and progression of cancer cells, and adipocytes are recognized as dominant actors for tumor progression. The soluble factors derived from adipocytes, called adipokines, including adiponectin, leptin, IL-6, and tumor necrosis factor α, are involved in tumor progression.

Adipocyte-derived leptin and IL-6 play key roles in the activation of the Jak/STAT and PI3K/Akt/mTOR pathways, which are frequently dysregulated in tumor pathogenesis. Recent studies have suggested that the PI3K/Akt/mTOR pathway is involved in adipokine-induced tumorigenesis [21].

The role of leptin in numerous neoplasms has been investigated. Chi et al. [43] proved that leptin secreted by adipocytes contributes to resistance to chemotherapy by melanoma cells. This resistance has been associated with an increased activation of the PI3K/Akt/mTOR and MEK/ERK survival pathways. The hyperactivation of these pathways has also been shown to impair the cytotoxic effect of 5-fluorouracil in colon cancer cells [44].

Another regulator of metabolic activity is AMPK (a protein kinase activated by adenosine monophosphate), which regulates the expression of IL-6 and IL-8 in adipocytes suggesting an involvement in the production of adipokines [45].

The energy state is signaled to the cell by mTOR through AMPK [46]. Several studies have revealed the convergence of the AMPK and mTOR signaling pathways, indicating that mTOR also plays a role as an integrator of signals derived from growth factors, nutrients, and cellular energy. AMPK and mTOR are therefore key factors in the adipocyte differentiation process; additional evidence shows that the expansion of adipose tissue in obesity is associated with a marked activation of mTOR, whereas the reduction of fat mass resulting from caloric restriction and fasting is associated with inhibition of mTOR in adipose tissue. Consequently, chronic pharmacological inhibition of the mTORC1 signaling pathway is associated with a reduction in adipose tissue as a result of the dimensional and numerical reduction of adipocytes. By exploiting the action of AMPK, cancer cells would obtain an advantage in growth and proliferation in conditions of nutrient and oxygen deprivation [45].

Fang et al. [46] also demonstrated that the duration of treatment with mTOR inhibitors can have different effects on metabolism, providing an explanation for the conflicting evidence. The authors showed that short-term treatment (2 weeks) with an mTOR inhibitor causes hyperlipidemia and insulin resistance and promotes hepatic gluconeogenesis, whereas prolonged treatment (20 weeks) results in reduced adiposity, increased insulin sensitivity, improved lipid profile, and higher energy expenditure [46].

In cancer patients, the measurement of skeletal muscle mass and adipose tissue therefore provides relevant information on body composition, nutritional status and the ability to metabolize drugs. Moreover, these elements play a key role in the pathogenesis and prognosis of oncological pathologies.

The measurement of body composition by CT scan has several advantages: non-invasiveness, high reproducibility and speed of execution.

These measurements can represent a sensitive and accurate estimate of the nutritional status allowing for quantifying the share of adipose tissue and skeletal muscle mass. An accurate determination of these indices can be an indicator of the nutritional status and biological factors that potentially influence the response to cancer treatments and the prognosis. The temporal monitoring of such measurements can be useful in better understanding biological mechanisms and possibly lead to the development of new therapeutic strategies. Few data are in fact present in the literature on the association of quantitative measures of body composition and survival.

The limitations of our study are mainly related to the retrospective design, which is not free from bias; other limitations include a small number of patients but still significant as regards rare neoplasms.

The impact of NET-associated fat and muscle loss cannot be precisely quantified because it is difficult to distinguish between causes of weight and mass loss. The duration of any treatment interruptions and associated toxicities varied between patients and could have influenced the results. Finally, the possible role of other confounding factors, such as concomitant therapies or comorbidities, cannot be excluded. A significant difference in OS is difficult to demonstrate in pre-treated patients and/or patients with advanced disease.

The measurements of skeletal muscle mass and adipose tissue in our study were made only at baseline and at the first radiological evaluation. Future studies could evaluate these variations for the entire duration of the treatment to better highlight correlations and/or changes in the results obtained throughout the treatment duration.

Despite the limitations, our study identified categories of patients who could benefit from everolimus treatment by identifying easily usable variables that could predict a benefit in PFS. Our work has also underlined the important role played by adipose and muscle tissue both in the pathogenesis and evolution of the disease and in the possible relationship to response to treatments.

## 5. Conclusions

In conclusion, this study showed that higher BMI, SMI and adipose tissue baseline values correlated with PFS in patients with NET who were treated with everolimus.

The mechanism is not well understood and our hypotheses need further confirmation in order to be validated. Despite the limitations described, our study aimed to investigate the role of adipose and muscle tissue in a setting of patients with rare cancer and how these could be related to the outcome. These indexes play an important role as indicators of the patient’s nutritional status but can be equally useful for evaluating association with the benefits derived from cancer treatment and survival. Future prospective studies on larger case series can be useful to validate our results.

## Figures and Tables

**Figure 1 cancers-14-03231-f001:**
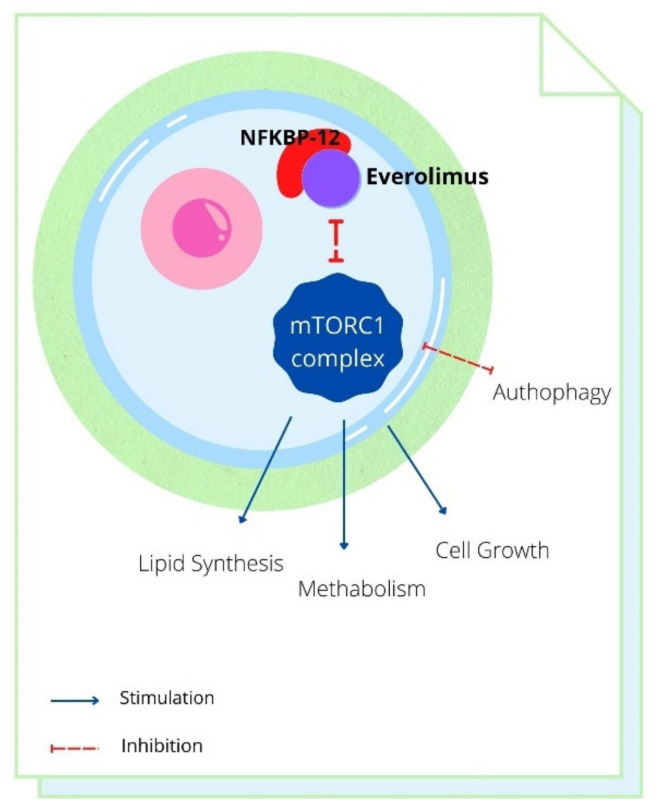
mTORC1 complex is involved in lipid synthesis and metabolism: Everolimus plus NFKBP-12 has a negative effect.

**Figure 2 cancers-14-03231-f002:**
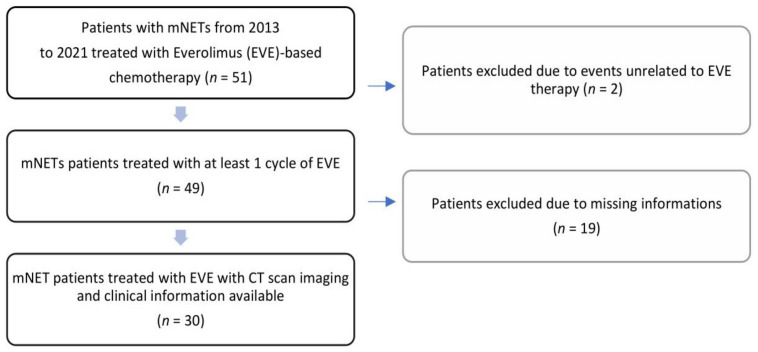
Flow diagram of patients identified and included in the final analysis. NET, neuroendocrine tumor; EVE, everolimus; CT, computed tomography.

**Figure 3 cancers-14-03231-f003:**
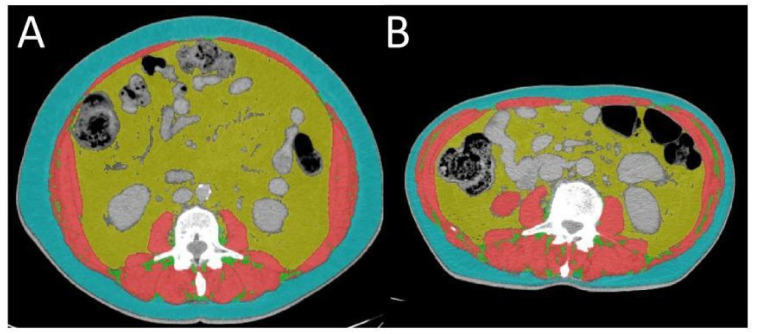
Evaluation of body composition. An axial CT image segmented into the skeletal muscle area (SKM in red), visceral adipose tissue area (VAT in yellow) and subcutaneous adipose tissue area (SAT in green). Total adipose tissue area (TAT) is identified from VAT + SAT. (**A**) and (**B**) differ from body composition.

**Figure 4 cancers-14-03231-f004:**
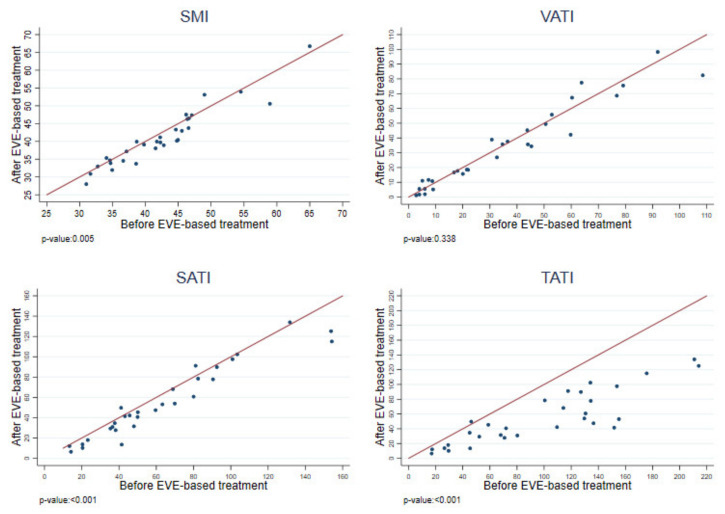
Indexes variation measured before and after the start of everolimus-based treatment.

**Figure 5 cancers-14-03231-f005:**
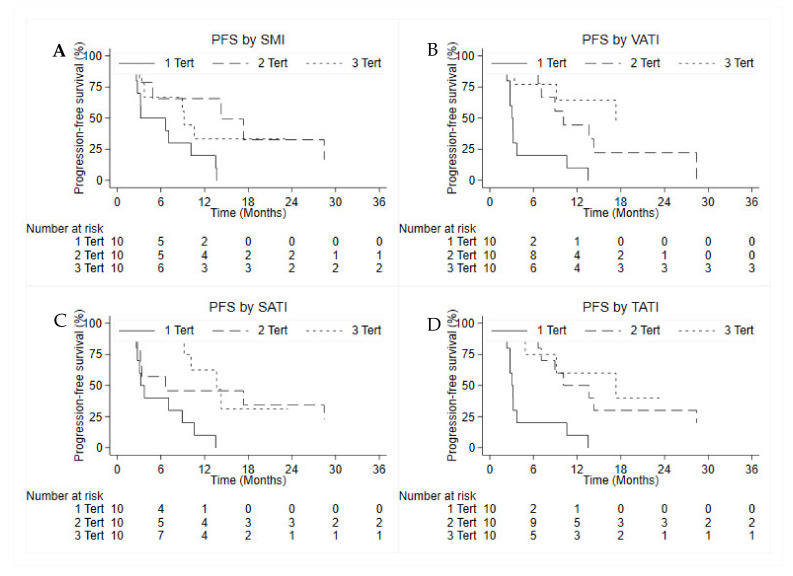
Correlation between PFS and SMI (**A**), VATI (**B**), SATI (**C**) and TATI (**D**).

**Figure 6 cancers-14-03231-f006:**
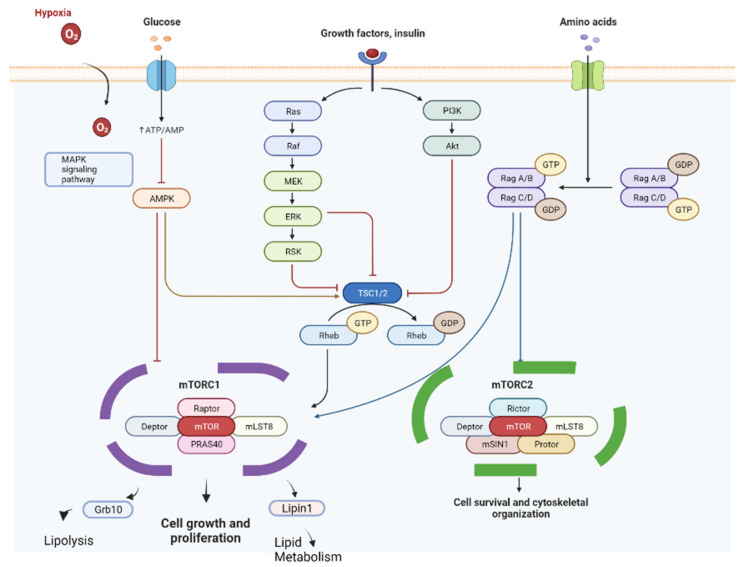
mTOR pathway is involved not only in tumor progression but also in lipid metabolism and lypolysis. Adapted from “mTOR Signaling Pathway”, by BioRender.com (2020). Retrieved from https://app.biorender.com/biorender-templates (accessed on 15 June 2022).

**Table 1 cancers-14-03231-t001:** Patient characteristics.

Characteristics	Median (Range)
Age at treatment	55.3 (24.6–70.2)
	**N (%)**
Gender	
Male	15 (50.0)
Female	15 (50.0)
Site of disease	
Pancreas	9 (30)
Gastrointestinal (GI)	12 (40)
Lung	6 (20)
Other	2 (6.7)
Unknown	1 (3.3)
Ki67	
≤10	18 (60)
>10	10 (33.3)
Unknown	2 (6.7)
Grading	
1	6 (20.0)
2	21 (70.0)
Unknown	3 (10)
Surgery of primitive	
Not done	11 (36.7)
Done	19 (63.3)
Comorbidity	
None	9 (30.0)
Only cardiovascular	8 (26.7)
Cardiovascular + Other (endocrine or metabolic)	8 (26.7)
Other	5 (16.7)
Previous treatments	
Somatostatin analogues (SSAs)	27 (90)
Peptide Receptor Radionuclide Therapy (PRRT) Chemotherapy Other (anti-VEGF agents)	25 (83.3) 16 (53.3) 2 (6.7)
Pre therapy BMI	
BMI ≤ 18.49	8 (26.7)
BMI > 18.49 and ≤ 24.99	18 (60.0)
BMI > 24.99	4 (13.3)

**Table 2 cancers-14-03231-t002:** Relation between best response and pre-therapy index variation.

Body Composition	Median (Range) for SD + PR Subgroup (*n* = 20)	Median (Range) for PD Subgroup (*n* = 10)	*p*-Value
SMI	42.5 (32.8–64.9)	39.2 (31.0–46.6)	0.186
VATI	40.2 (3.9–108.5)	7.4 (2.8–76.8)	0.015
SATI	66.6 (13.4–154.2)	41.3 (14.2–103.4)	0.086
TATI	122.4 (17.4–214.1)	49.2 (17.0–136.4)	0.027

**Table 3 cancers-14-03231-t003:** Relation among best response and index variation.

Variation of Body Composition from Pre Therapy to Post Therapy	SD + PR Subgroup (n = 20) (%)	PD Subgroup (n = 10) (%)	*p*-Value
**SMI variation**			
Increase	9 (45.0)	0 (0.0)	0.011
Decrease	11 (55.0)	10 (100.0)	
**VATI variation**			
Increase	9 (45.0)	3 (30.0)	0.429
Decrease	11 (55.0)	7 (70.0)	
**SATI variation**			
Increase	2 (10.0)	1 (10.0)	
Decrease	18 (90.0)	9 (90.0)	1.000
**TATI variation**			
Increase	0 (0.0)	1 (10.0)	0.150
Decrease	20 (100.0)	9 (90.0)	

**Table 4 cancers-14-03231-t004:** Progression-free survival by subgroups.

Variables		N. pts	N. Events	Median PFS (95%CI)	*p*-Value
**Total**		30	23	8.9 (3.4–13.7)	-
Age at treatment	<55 years	14	12	7.1 (2.7–13.5)	0.695
	≥55 years	16	11	9.2 (3.2–17.3)	
Gender	Male	15	11	9.1 (3.2–13.7)	0.875
	Female	15	12	8.9 (2.8–28.4)	
Site of disease	Pancreas	9	7		
	Gastro-intestinal	12	8		
	Lung	6	5		
	Other	2	2		
Previous surgery	No	11	8	8.9 (2.8–41.69	0.262
	Yes	19	15	6.6 (3.1–14.2)	
Ki-67	Ki67 ≤ 10	18	13	10.6 (2.8–14.3)	0.251
	Ki67 > 10	10	8	7.1 (3.2–10.1)	
Grading	G1	5	5	10.1 (3.2-NE)	0.206
	G2	19	12	13.7 (3.2–41.6)	
BMI	≤18.49	8	7	3.2 (0.9–6.7)	0.011 ^#^
	>18.49 and ≤24.99	18	14	10.1 (3.7–28.4)	
	>24.99	4	2	-	
SMI	1 tertile	10	10	3.2 (0.9–10.1)	0.039
	2 tertile	10	6	14.2 (2.3-NE)	
	3 tertile	10	7	9.1 (2.7-NE)	
VATI	1 tertile	10	10	3.1 (0.8–3.7)	<0.001
	2 tertile	10	8	10.1 (4.9-NE)	
	3 tertile	10	5	17.3 (2.6-NE)	
SATI	1 tertile	10	10	3.2 (2.3–8.9)	0.014
	2 tertile	10	7	6.6 (0.9-NE)	
	3 tertile	10	6	13.6 (4.8-NE)	
TATI	1 tertile	10	10	3.1 (0.9–3.7)	<0.001
	2 tertile	10	8	10.1 (3.4–28.4)	
	3 tertile	10	5	17.3 (2.6-NE)	

^#^ log-rank test calculated only between BMI ≤ 18.49 and BMI > 18.49 and ≤24.99 subgroups. NE—Not estimable from statistical software.

**Table 5 cancers-14-03231-t005:** Evaluation of toxicities.

Toxicity	G1	G2	G3	G4
Mucositis	2 (6.7)	4 (13.3)	5 (16.7)	0 (0.0)
Pulmonary toxicity	1 (3.3)	5 (16.7)	0 (0.0)	0 (0.0)
Hypercreatininemia	2 (6.7)	1 (3.3)	0 (0.0)	0 (0.0)
Thrombocytopenia	2 (6.7)	4 (13.3)	1 (3.3)	0 (0.0)
Skin toxicity	4 (13.3)	4 (13.3)	1 (3.3)	0 (0.0)
Fatigue	1 (3.0)	6 (20.0)	1 (3.3)	0 (0.0)
Hyperglycemia	5 (16.7)	3 (10.0)	0 (0.0)	0 (0.0)
Neutropenia	3 (10.0)	4 (13.3)	0 (0.0)	0 (0.0)
Liver toxicity	3 (10.0)	1 (3.3)	3 (10.0)	0 (0.0)
Hypertriglyceridemia	2 (6.7)	2 (6.7)	0 (0.0)	0 (0.0)
Gastrointestinal toxicity	4 (13.3)	3 (10.0)	1 (3.3)	0 (0.0)

**Table 6 cancers-14-03231-t006:** Toxicities by tertiles.

Tertiles of Body Composition		No Tox or G1–G2	G3	*p*-Value
SMI	1 tertile	8 (40.0)	2 (20.0)	0.549
	2 tertile	6 (30.0)	4 (40.0)	
	3 tertile	6 (30.0)	4 (40.0)	
VATI	1 tertile	8 (40.0)	2 (20.0)	0.549
	2 tertile	6 (30.0)	4 (40.0)	
	3 tertile	6 (30.0)	4 (40.0)	
SATI	1 tertile	10 (50.0)	0 (0.0)	0.024
	2 tertile	5 (25.0)	5 (50.0)	
	3 tertile	5 (25.0)	5 (50.0)	
TATI	1 tertile	8 (40.0)	2 (20.0)	0.350
	2 tertile	7 (35.0)	3 (30.0)	
	3 tertile	5 (25.0)	5 (50.0)	

## Data Availability

The datasets generated and/or analyzed during the current study are available from the corresponding author on reasonable request.

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
