# Peer review of "Prognostic and Predictive Role of Body Composition in Metastatic Neuroendocrine Tumor Patients Treated with Everolimus: A Real-World Data Analysis"

_cancers, 2022, doi:10.3390/cancers14133231_

Round 1

Reviewer 1 Report

the Authors have conducted an interesting study to investigate the role of body composition indexes in patients with metastatic NETs treated with everolimus. 

the methods of the study are very sound and the results have potential relevance for clinical practice. However, I feel that the quality of presentation should be improved.

For instance, the Introduction is very long, and I think that the first 3-4 paragraphs can be easily shortened, or even better omitted. The same applies to the first paragraphs of the Discussion.

Furthermore, patients selection (see patients and methods section) is hard to follow. the same information is repeated in the Results.

the above are only examples of suboptimal reporting of the entire paper. can the Authors make an effort to improve presentation?

thanks

Author Response

We are returning the above manuscript, revised in accordance with reviewers’ indications,

for further evaluation. All changes have been underlined and highlighted in red. A point-by-

point reply to the reviewers’ comments is also enclosed.

Comments and Suggestions for Authors

the Authors have conducted an interesting study to investigate the role of body composition indexes in patients with metastatic NETs treated with everolimus. 

the methods of the study are very sound and the results have potential relevance for clinical practice. However, I feel that the quality of presentation should be improved.

For instance, the Introduction is very long, and I think that the first 3-4 paragraphs can be easily shortened, or even better omitted. The same applies to the first paragraphs of the Discussion.

Furthermore, patients selection (see patients and methods section) is hard to follow. the same information is repeated in the Results.

Reply: we thank the reviewer for this point of view. We improve the flow diagram in order to be more easy the understanding of patients’ selection.

the above are only examples of suboptimal reporting of the entire paper. can the Authors make an effort to improve presentation?

thanks

Reply: We thank the reviewer for positive comments and to give us a different point of view.  We revise both the introduction and discussion removing redundant phrases.

Furthermore we add 2 figures (as requested also by the  reviewer 2) to better explain the role of mTOR pathway. Furthermore an improvement throughout the paper has been done.

English language was revised by a professional editing language service.

An update of the references has been done

Reviewer 2 Report

Title: Prognostic and Predictive role of Body Composition in Metastatic Neuroendocrine Tumor patients treated with Everolimus: a real world data

Manuscript ID: 1725884

The authors have perfectly presented the data for demonstrating a correlation between body mass compositions and metastatic neuroendocrine tumor of patients. For conducting this research, the authors included 30 patients with  neuroendocrine tumor whome were treated with everolimus. The body composition indexes (skeletal muscle index [SMI] and adipose tissue indexes) were assessed by measuring on a computed tomography (CT) scan the cross-sectional area at L3 at baseline and at the first radiological assessment after start of treatment. The body mass index (BMI) was assessed at baseline. The authors discovered that; the median progression-free survival (PFS) was 8.9 months. Similarly, the other body composition indexes also showed a statistically significant difference in the three groups based on tertiles. The median PFS was 3.2 months in underweight patients and 10.1 months in normal-weight patients. There were no significant differences in terms of overall survival. The authors concluded that; there is a correlation between PFS and the body composition indexes in patients with NETs treated with everolimus, underlining the role of adipose and muscle tissue in patients NETs.

Although this study is perfectly drafted, reviewed and presented, the reviewer would like to suggest the following minor and major comments to further improve the quality of the manuscript.

Major comments

1. Add a figure/flowchart describing the intracellular pathway presented in the discussion section.

2. A pathway diagram showing the mTOR pathway can be added.

Minor comments:

1. Abstract: Background: Neuroendocrine tumors (NETs) are rare neoplasms frequently characterized Of by

2.   . 1 Introduction -Remove the dot here. Same for the results and discussion section

3. In the discussion section, correct –“and adipocytes are recognized as dominant actors” Is it actors of factors?

4. More than 80% of the references should be less than 5-years old. Please change the references to the most recent ones.

In the title of the manuscript, see id real world is written as real-world or real world?

Author Response

Reply to reviewer 2

We thank the reviewer 2 for the positive feedback. We modify the text according to suggestions. All the modifications are in red.

Manuscript ID: 1725884

The authors have perfectly presented the data for demonstrating a correlation between body mass compositions and metastatic neuroendocrine tumor of patients. For conducting this research, the authors included 30 patients with  neuroendocrine tumor whome were treated with everolimus. The body composition indexes (skeletal muscle index [SMI] and adipose tissue indexes) were assessed by measuring on a computed tomography (CT) scan the cross-sectional area at L3 at baseline and at the first radiological assessment after start of treatment. The body mass index (BMI) was assessed at baseline. The authors discovered that; the median progression-free survival (PFS) was 8.9 months. Similarly, the other body composition indexes also showed a statistically significant difference in the three groups based on tertiles. The median PFS was 3.2 months in underweight patients and 10.1 months in normal-weight patients. There were no significant differences in terms of overall survival. The authors concluded that; there is a correlation between PFS and the body composition indexes in patients with NETs treated with everolimus, underlining the role of adipose and muscle tissue in patients NETs.

Although this study is perfectly drafted, reviewed and presented, the reviewer would like to suggest the following minor and major comments to further improve the quality of the manuscript.

Major comments

  1. Add a figure/flowchart describing the intracellular pathway presented in the discussion section.
  2. A pathway diagram showing the mTOR pathway can be added.

Reply: we thank the reviewer for the opportunity to improve the manuscript. A figure (Fig. 4) was added to explain the intracellular mTOR pathway and a more schematic one about the role of  mTORC 1 (fig.1)

Minor comments:

  1. Abstract: Background: Neuroendocrine tumors (NETs) are rare neoplasms frequently characterized Of by

Reply: we correct the phase.

  1.   . 1 Introduction -Remove the dot here. Same for the results and discussion section

Reply: we remove dot in both sections

  1. In the discussion section, correct –“and adipocytes are recognized as dominant actors” Is it actors of factors?
  2. More than 80% of the references should be less than 5-years old. Please change the references to the most recent ones.

reply: We agree the reviewer . An extensive update of the literature has been done

In the title of the manuscript, see id real world is written as real-world or real world? 

Reply: we thank the reviewer . We have corrected  the title. Page 1 line 2

Round 2

Reviewer 1 Report

thanks for considering my comments. I have just one further: the Authors claim that a professional editing service has been used. However, I can't find it cited in the acknowledgments, along with the funding source.

this is not in line with current Good Publication Practice and therefore the paper cannot be published without this information. can the Authors revise?

Author Response

We are returning the above manuscript, revised in accordance with reviewer’s indications, for further evaluation.

Reply: We agree the reviewer and we add a sentence about the service used in the acknowledgement section, highlighted in green